# Mapping HIV-1 RNA structure, homodimers, long-range interactions and persistent domains by HiCapR

Yan Zhang[1†], Jingwan Han[2†], Xie Dejian[1†], Wenlong Shen[1], Ping Li[1], Jian You Lau[3], Jingyun Li[2*], Lin Li[2*], Grzegorz Kudla[3*], Zhihu Zhao[1*]

[1]Laboratory of Advanced Biotechnology, Beijing Institute of Biotechnology, Beijing, China; [2]Institute of Microbiology and Epidemiology, Beijing, China; [3]MRC Human Genetics Unit, University of Edinburgh, Edinburgh, United Kingdom

**\*For correspondence:**
lijyjk@163.com (JL);
dearwood@sina.com (LL);
gkudla@gmail.com (GK);
zhaozh@bmi.ac.cn (ZZ)

[†]These authors contributed equally to this work

## eLife Assessment

This manuscript focuses on the identification of RNA crosslinks within the HIV RNA genome under different conditions i.e. in infected cells and in virions using a new method called HiCapR. These cross-links reveal long-range interactions that can be used to determine the structural arrangement of the viral RNA, providing **valuable** data that show differences in the genomic organization in different conditions. The data analysis, however, is **incomplete** and based on extensive computational analysis from a limited number of datasets.

**Abstract** Human Immunodeficiency Virus type 1 (HIV-1) RNA genome organization remains a critical knowledge gap in understanding its replication cycle. To address this, we developed HiCapR, a psoralen crosslinking-based RNA proximity ligation method coupled with post-library hybridization, enabling high-resolution mapping of RNA-RNA interactions across the HIV-1 genome. This approach confirmed canonical structural motifs, including stem-loop architectures in the 5'-untranslated region (5'-UTR) and Rev Response Element (RRE), as well as dimerization sites within the 5'-UTR critical for viral packaging. Notably, HiCapR identified novel homodimerization events distributed along the genome, suggesting an expanded regulatory role of RNA multimerization in splicing regulation and selective encapsidation. Intriguingly, while infected cells exhibited extensive long-range RNA interactions—particularly within the 5'-UTR—virion-packaged genomes displayed a marked reduction in such interactions, indicative of a structural transition from a loosely organized state to a condensed conformation. This spatial reorganization coincided with the preservation of stable genomic domains essential for dimerization, which persisted throughout virion assembly. These domains, enriched at homodimer interfaces, likely serve as structural scaffolds ensuring fidelity during genome packaging. This work establishes HiCapR as a robust tool for probing RNA interactomes and provides mechanistic insights into how HIV-1 exploits RNA topological heterogeneity to regulate its life cycle. The identification of conserved structural domains and transient interaction networks opens avenues for targeting RNA conformation in antiviral strategies.

## Introduction

Human Immunodeficiency Virus (HIV) remains a significant global health concern despite advancements in antiretroviral therapy even with the newest report of total seven AIDS patients been completely 'cued' (*Mallapaty, 2024*; *Freed, 2015*; *Emery and Swanstrom, 2021*). Research on HIV-1 RNA structure and function has intensified, focusing on assembly, release, and maturation processes,

using NMR (*Keane et al., 2015*), cryo-EM (*Zhang et al., 2018*), and chemical probing (*Watts et al., 2009*; *Tomezsko et al., 2020*; *Ye et al., 2022*). However, our knowledge of the global architecture and the long-range interactions within the HIV-1 RNA genome remains limited, primarily due to the complexity and dynamic nature of these structures.

The HIV genome, comprising two copies of positive-sense RNA, has been extensively studied to understand its regulatory elements, such as the 5'-UTR, crucial for dimerization and selective packaging into viral particles (*Ding et al., 2021*). Additionally, interactions involving the nucleocapsid protein (NC) and the dimerization signal (Ψ) play pivotal roles in the assembly and maturation of HIV virions (*Chen et al., 2020*). Furthermore, the RRE in the HIV genome, responsible for nucleocytoplasmic transport of viral RNA, exhibits structural variability impacting virus replication rates (*Sherpa et al., 2015*; *Jackson et al., 2016*). While current techniques like SHAPE began to provide insights into local RNA structures (*Tomezsko et al., 2020*; *Ye et al., 2022*; *Saldi et al., 2021*), studying the intricate long-range RNA interactions and homodimers within the HIV genome under physiological conditions remains a significant challenge, primarily due to the limitations of these methods in capturing the full spectrum of RNA interactions and the dynamic nature of the viral genome.

Proximity ligation-based methods have been shown to be particularly useful for studying long-range RNA interactions, as they can overcome the RNA length limitations of traditional techniques. These methods have been applied to the study of various viruses, including influenza (*Le Sage et al., 2020*; *Miyamoto and Noda, 2020*), Zika (*Li et al., 2018*; *Ziv et al., 2018*; *Huber et al., 2019*), and SARS-CoV-2 (*Ziv et al., 2020*; *Cao et al., 2021*; *Yang et al., 2021*; *Zhang et al., 2021*), and have led to the identification of numerous RNA structures that are closely related to the virus lifecycle. We and others have recently shown that RNA proximity ligation data also contains information about RNA homodimerization events (*Gabryelska et al., 2022*; *Zhang et al., 2022*). Nevertheless, as of now, there are no proximity ligation studies of long-range interactions or dimeric interactions of the HIV RNA genome. Probably due to high requirements of starting material, making it very difficult for low abundant HIV sample treatment.

In this study, we modified our previous protocol (*Zhang et al., 2021*) by integrating a hybridization-based capture of HIV-1 sequence after library construction, resulting in a method we call High throughput Capture of RNA interactions (HiCapR). This advancement enabled the comprehensive capture and analysis of the complete HIV RNA genome. By applying HiCapR to both infected cells and virions, we uncover a distinct genomic compression pattern in virions, highlighting the critical role of global genome folding in HIV packaging and assembly. Our findings not only reveal the presence of persistent genomic domains within virions that facilitate whole genome dimerization, even the transgenerational inheritance but also a significant reduction of long-range RNA-RNA interactions in virions compared to infected cells. These insights provide a new perspective on the functional implications of HIV RNA structure in splicing and packaging processes and may lead to the identification of new targets for antiviral intervention.

## Results

### Capturing in vivo HIV genome RNA structure in infected cells and in virion

To effectively capture and analyze full-length low abundant HIV RNA-RNA interactions in infected cells, we developed an improved version of the protocol, which we call *HiCapR*. This method incorporates hybridization-based capture with next-generation sequencing (NGS) library construction, a principle well established in capture Hi-C methods (*Jäger et al., 2015*; *Dryden et al., 2014*; *Sahlén et al., 2015*; *Mifsud et al., 2015*) and in RNA proximity ligation (*Ziv et al., 2018*; *Ziv et al., 2020*). Applying this strategy, we compared the NL4-3 and GX2005002 strains *Chang et al., 2015*,which are two distinct HIV-1 strains that exhibit different prevalence patterns in various geographical regions. The NL4-3 strain is a well-characterized laboratory strain that is widely used in HIV research and is representative of the HIV-1 subtype B, which is highly prevalent in Europe and the Americas. On the other hand, GX2005002 is a primary isolate of the CRF01_AE subtype, which is one of the most prevalent strains in Southeast Asia, particularly in China. The overall design of this study is presented in *Figure 1A*, with detailed methods provided in the Methods section. *Figure 1—figure supplement 1* illustrates the bioinformatic pipeline used to detect RNA-RNA interactions and homodimers.

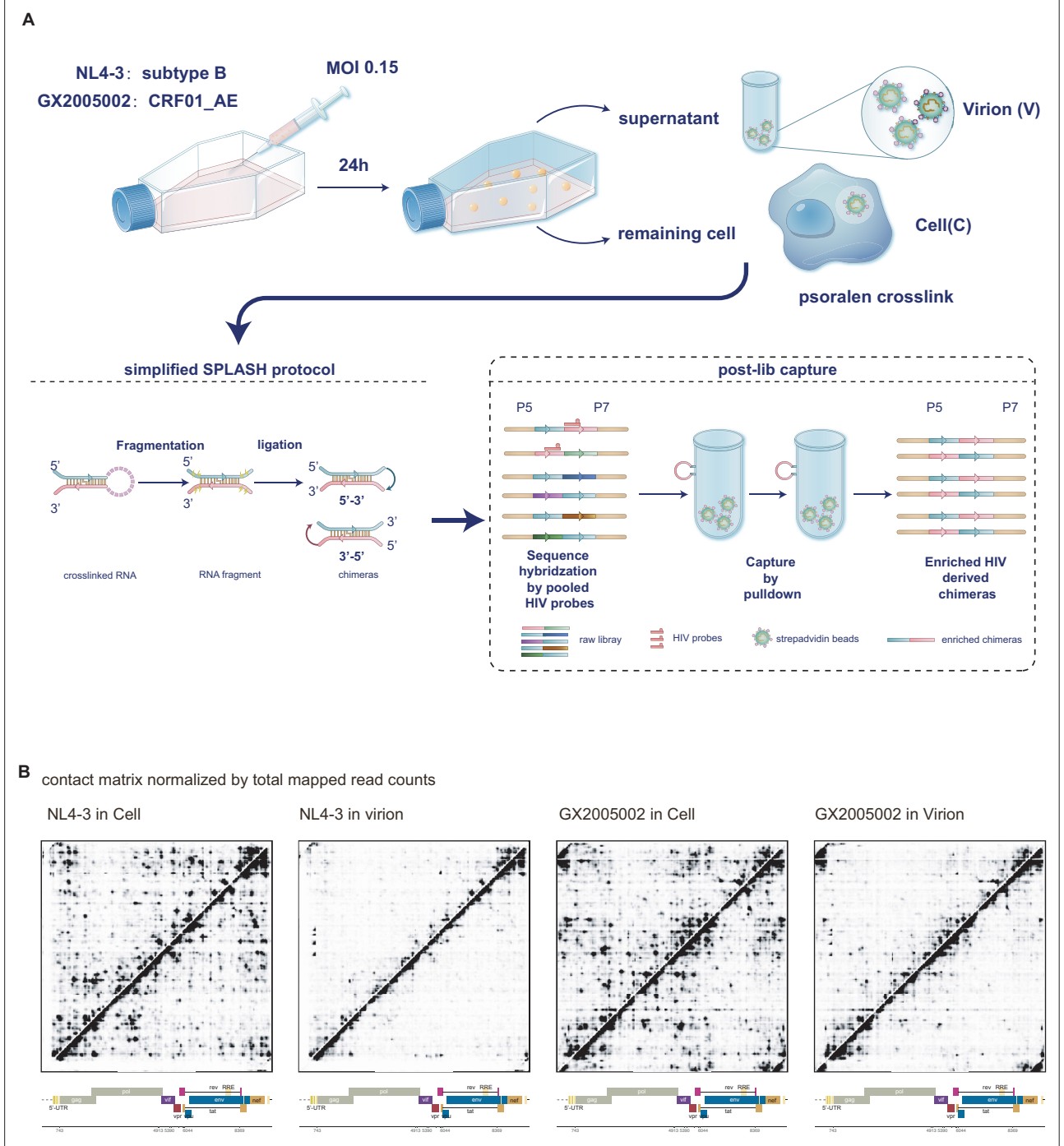

**Figure 1.** Experimental design of HiCapR and overall HIV-1 contact matrix. (**A**) Experimental design of HiCapR for profiling RNA-RNA interactions and dynamics of the HIV genome. Based on the simplified SPLASH protocol, HiCapR incorporates a post-library probe-based hybridization and streptavidin pulldown method to enrich HIV RNA chimeras from the SPLASH library. The HiCapR method has been applied to two strains of HIV, NL4-3 and GX2005002, in both infected cells and their corresponding virus particles. (**B**) Comprehensive contact matrix derived from infected cells and virions of both NL4-3 and GX2005002 HIV-1 strains in the HiCapR experimental groups. The heatmap displays the average count of chimeras per 1 million mapped reads, combining data from two biological replicates for each sample.

The online version of this article includes the following source data and figure supplement(s) for figure 1:

**Source data 1.** Designed probes for enriching HIV chimeras in fasta format.

**Figure supplement 1.** Workflow for identifying chimeric reads supporting RNA-RNA interactions as well as homodimers.

**Figure supplement 2.** HIV genome coverage of HiCapR data.

*Figure 1 continued on next page*

*Figure 1 continued*

**Figure supplement 3.** Sample correlation.

**Figure supplement 4.** Contact matrix derived from infected cells and virions of both NL4-3 and GX2005002 HIV-1 strains.

Approximately 16 million raw reads were obtained per sequencing library, with over 95% of the reads containing HIV sequences, indicating efficient capture of HIV RNA transcripts. The alignment rate exceeded 90%, and we achieved an average depth of coverage of approximately 11,500 X across the genome (*Supplementary file 1*). The uniform distribution of reads post-capture (*Figure 1—figure supplement 2*) demonstrates HiCapR's effectiveness in capturing the entire HIV RNA genome with minimal bias. The overall duplication rate remained below 50%, with a lower rate in the ligation group, suggesting increased library complexity due to RNA proximity ligation.

Chimeric reads were identified using the hyb technique as previously described (*Zhang et al., 2021*), with a database comprising human spliced mRNAs, noncoding RNAs (*Travis et al., 2014*) and the HIV genome sequence as reference. The chimeric rate, a crucial indicator of ligation efficiency, was approximately 1% in non-ligation controls and around 9% in the ligation group, aligning with the widely reported range. The proportion of 5'–3' and 3'–5' chimeras was nearly equal in long-range interactions, while proximal interactions were enriched for 3'–5' chimeras. Pearson correlation analysis revealed a high degree of correlation ($r>0.99$) between biological replicates, indicating high reproducibility. Additionally, a significant similarity was observed between virions in the supernatant and infected cells from the same viral strain (*Figure 1—figure supplement 3*).

Using a similar approach as described in our previous paper (*Zhang et al., 2021*), we generated a contact matrix for the global HIV-1 genome, revealing clear local and long-range interactions (*Figure 1B*, *Figure 1—figure supplement 4*). In summary, HiCapR demonstrated high reproducibility, low bias, reliability, and efficiency in capturing HIV transcripts, making it a robust tool for analyzing the RNA structure of the HIV genome and potentially other similar RNA viruses.

## Local RNA structure heterogeneity, dynamics and robustness in HIV 5'-UTR

We aimed to characterize specific local structures and their dynamics within the HIV genome. One of the extensively investigated structures is 5'-UTR, as these structures are closely associated with crucial life processes of HIV.

Chimeras analysis support the presence of key structural elements, including TAR, polyA, SL1, SL2, and SL3, as well as polyA-SL1 in the monomeric conformation in the HIV genome (*Figure 2A*, *Figure 2—figure supplement 1*). Despite the pivotal role of the 5'-UTR in replication, packaging, transcription and translation, we noted that the 5'-UTR sequence of HIV is not highly conserved across different HIV strains. A comparison between the NL4-3 and GX2005002 strains revealed notable insertions and deletions in the U5 region, along with point mutations in the loop of the SL1 core region, which is critical for HIV dimerization (*Figure 2B*). These findings raise questions about the conservation of the structural integrity of this region. We therefore applied the comradesFold algorithm (*Ziv et al., 2018*) to our HiCapR data to generate 1000 structure predictions of the 5'-UTR (extending 100 nt downstream of the AUG start codon) for each sample, and then performed MDS analysis after aligning the viral genome coordinates of both strains (*Figure 2C*). This analysis revealed that the reported dimer and monomer structures clustered together in NL4-3 but form distinct clusters in GX2005002 (*Figure 2C*). Based on these folded structures, we calculated the base pairing probability for each base pair, which involves determining the number of folded structures supporting a specific base pair divided by the total number of structures. Visualizing this base pairing probability as a heatmap identifies the most stable base pairs in the 5' UTR of HIV. We observed a consistent presence of key structural elements such as polyA, TAR, SL1, SL2, and SL3 in both NL4-3 and GX2005002 strains (*Figure 2D*), suggesting robustness in the overall structure despite sequence variations and alternative RNA conformations.

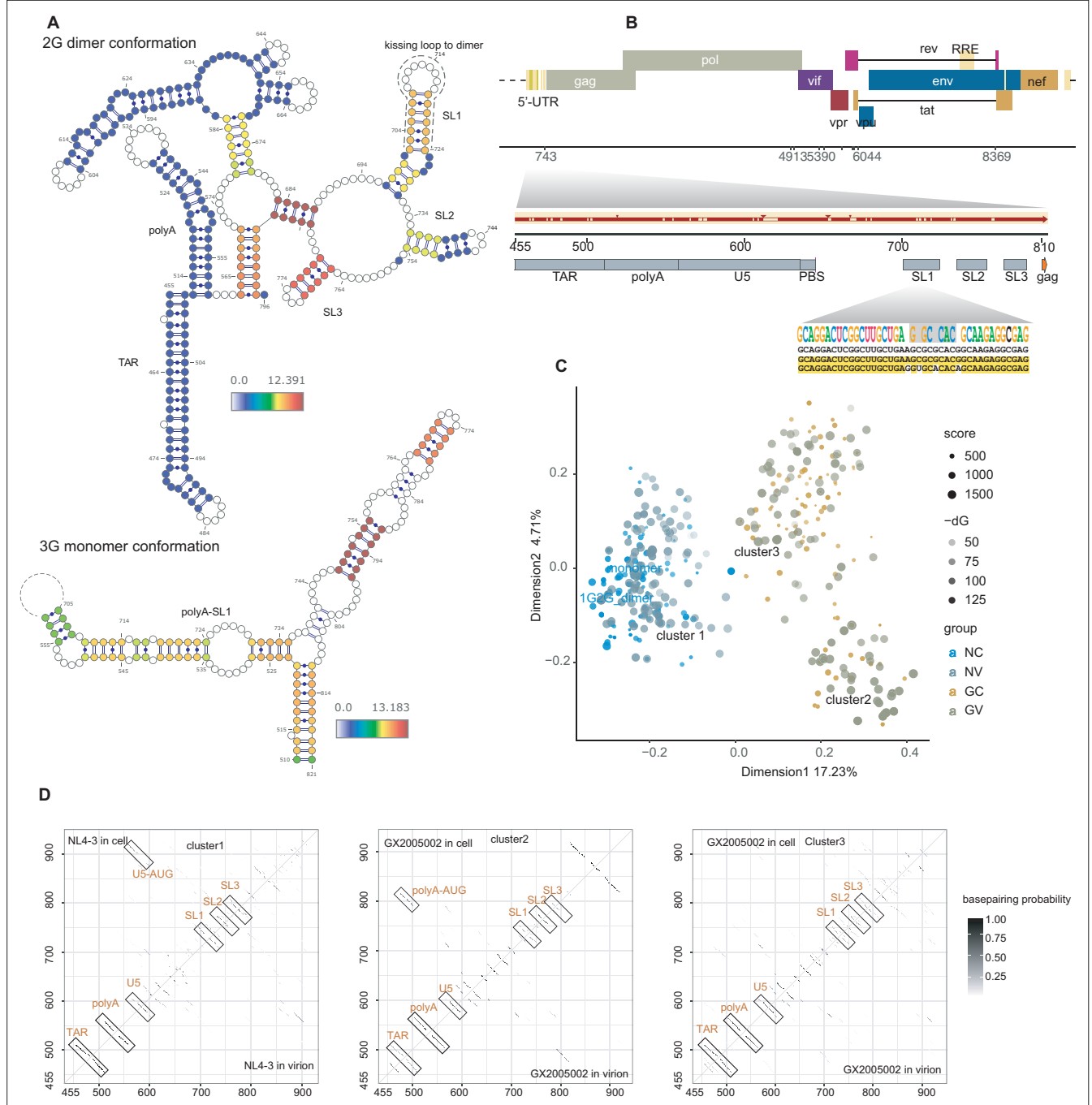

**Figure 2.** conformations of HIV-1 5'-UTR. (**A**) Known conformation of HIV-1 5'-UTR supported by chimeras. Previous reported stems in 5'-UTR are supported by chimeras from HiCapR. Colors of the nucleotides indicate the log2 transformed base pairing scores. (**B**) Representation of the one-dimensional structure of the HIV-1 5'-UTR, highlighting the conservation between the GX2005002 and NL4-3 strains in this region. The diagram includes rectangular boxes denoting the locations of key structural elements, with numerical coordinates referencing NCBI DNA genome coordinates. Dashed boxes indicate regions that are either absent or distinct in GX2005002 compared to NL4-3. Small triangle arrows indicate insertions in the GX2005002 5'-UTR. At the top, a seqlogo displays the consensus nucleotides in the SL1 region. (**C**) Multidimensional scaling (MDS) plot clustering each of the 1,000 computationally predicted structures of the 5'-UTR in two strains under two conditions. (**D**) Basepairing probability matrices were calculated from 1000 computationally predicted structures, with color indicating the percentage of structures supporting the specific basepair.

The online version of this article includes the following figure supplement(s) for figure 2:

**Figure supplement 1.** Dimer and monomer conformations of HIV-1 5'-UTR.

## Novel homodimerization sites and their implications in RNA splicing and viral assembly

Proximity ligation techniques enable unambiguous detection of RNA homodimers from 'overlapping' chimeric reads (*Gabryelska et al., 2022*; *Zhang et al., 2022*; *Figure 3A*). To ensure the accuracy of our results, we implemented a rigorous data filtering process to select chimeras formed exclusively through dimerization, minimizing background interference caused by RNA interactions or self-ligation (*Figure 1—figure supplement 1* and Methods). We quantified the total number of reads for dimeric chimeras, 3'–5' chimeras, and 5'–3' chimeras respectively in each sample (*Figure 3—figure supplement 1*) and plotted contact matrices made from non-overlapping chimeras and dimeric chimeras (*Figure 3—figure supplement 2*). The non-overlapping chimeras reveal short-range intramolecular RNA-RNA interactions, while dimeric chimeras capture homodimer formation in the HIV genome.

The strongest homodimerization signal was found in the 5'-UTR region, which aligns with previous studies. The 5'-UTR region is well-known for its role in triggering HIV dimerization. Previous literature has highlighted the importance of SL1 (DIS) and SL3 (Ψ) in 5'-UTR (*Ye et al., 2022*; *Stephenson et al., 2013*; *Mailler et al., 2016*). Our data shows that the SL1, SL2, and SL3 regions all have supportive dimeric chimeras (*Figure 3—figure supplement 2*). We derived a Dimerization Score by calculating the reads that support base pairing of homodimeric chimeras (*Figure 3—figure supplement 3*), analogous to the COMRADES Score described previously (*Ziv et al., 2018*; *Ziv et al., 2020*; *Zhang et al., 2021*). The base pairing of homodimers in the SL1 region of the HIV genome is consistent with previous data from NMR and cyro-EM studies *Ding et al., 2020*. Additionally, we observed that variations in nucleotides within the SL1 region do not alter the base pairing pattern of dimers between the NL4-3 and GX2005002 strains (*Figure 3B*) which is egret with previous structure robustness revealed by permutation and structure prediction results.

Strikingly, in addition to the known homodimer within the SL1 and SL3 regions, we observed homodimerization distributed along the entire length of the HIV genome (*Figure 3—figure supplement 4*). Homodimers were present in both infected cells and virions of both strains, with approximately 20 peaks of dimerization that are conserved between the NL4-3 and GX2005002 strains along the genome (*Supplementary file 2*).

We investigated whether dimers could be predicted by the strength of base pairing of intermolecular loop-loop interactions by performing a systematic molecular hybridization analysis on the genomes of two HIV-1 strains, NL4-3 and GX2005002. Our findings showed no correlation between the predicted folding energy and the abundance of measured dimeric chimeras (*Figure 3—figure supplement 4A*), suggesting that the formation of HIV homodimers is not solely a consequence of local base pairing propensities. Instead, it implies that additional factors, such as the binding of specific proteins, may significantly influence the dimerization process, potentially playing a more decisive role in stabilizing or facilitating the formation of these homodimers.

The identification of multiple dimers in HIV RNA beyond the 5'-UTR region raises intriguing questions regarding their potential roles serving within the HIV genome. To this end, we analyzed the sequence within dimer peaks, identifying a prevalent AG-rich motif rich, present in nearly every peak. This motif highly? resembles the RNA binding motif of Serine/Arginine-Rich Splicing Factor (SRSF; *Figure 3—figure supplement 4B, C*; *Figure 3—figure supplement 5*), which are essential RNA-binding proteins involved in pre-mRNA splicing and alternative splicing regulation (*Änkö et al., 2012*), serving as the main type of HIV-1 splicing factors (*Sertznig et al., 2018*). This result suggests a potential role of dimerization in RNA splicing processes within the HIV genome.

5'–3' discontinuous reads were identified in non-ligation control data, enabling the inference of splicing junction sites. These inferred sites showed strong concordance with canonical splicing sites and recent nanopore sequencing data (*Nguyen Quang et al., 2020*; *Figure 3B*). Notably, almost every junction site, including splicing donor and acceptor sites, exhibited dimer peaks around them. Additionally, the majority of dimeric chimeras covered these junction sites and enriched downstream of acceptor sites (*Figure 3—figure supplement 5*), suggesting that dimerization process/events takes place on unspliced genomic RNA. Interestingly, dimers surrounding splicing regulatory elements (as summarized in *Sertznig et al., 2018*) form stable base pairing supported by numerous chimeras (*Figure 3—figure supplement 5*). This observation underscores the close proximity and potential functional relationship between dimerization events and RNA splicing processes at these critical sites

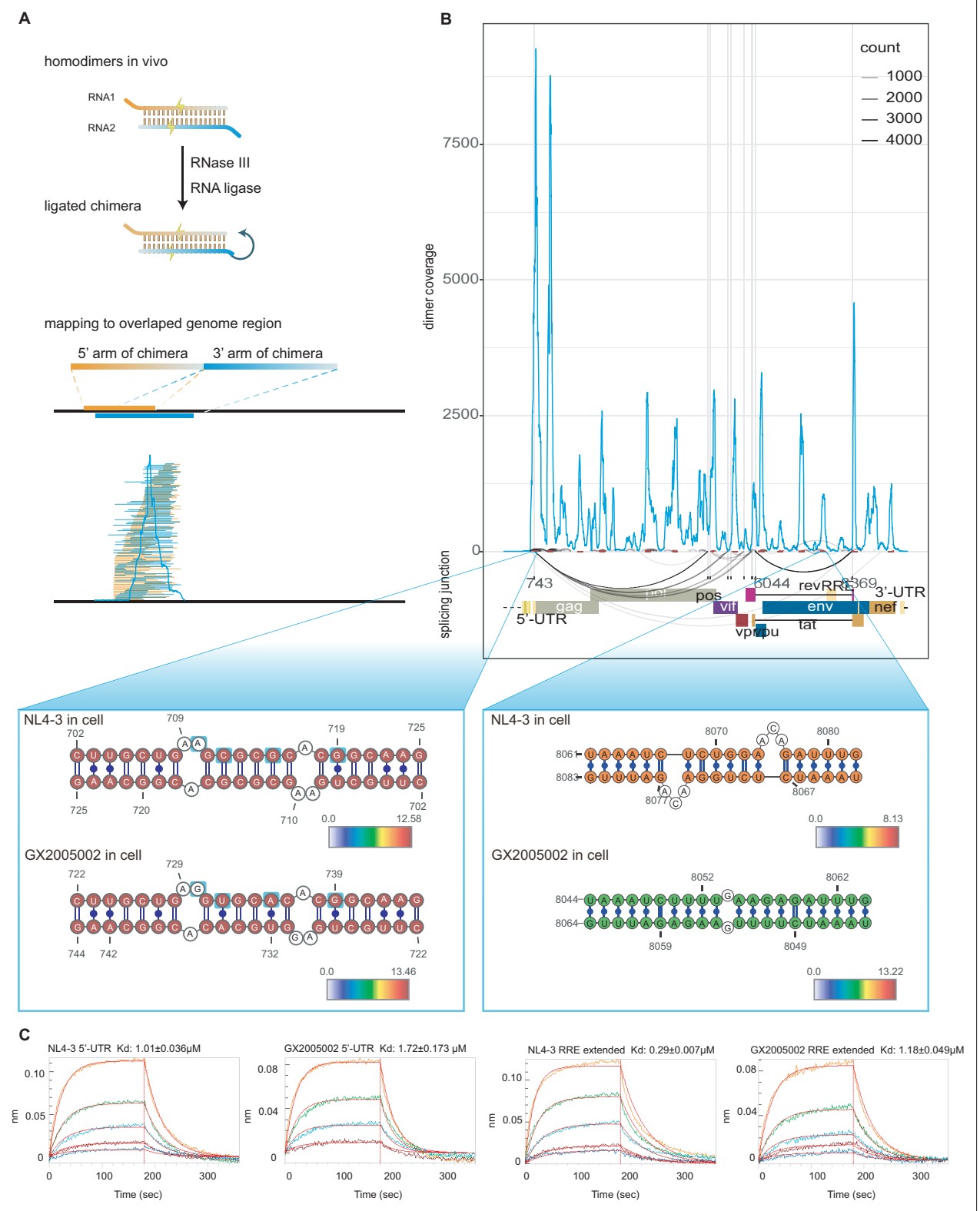

**Figure 3.** Identification and validation of homodimers in HIV-1 genome. (**A**) Visual depiction of the RNA homodimer formation process: Inter-ligated fragments (arms) originating from homodimeric RNA molecules generate chimeras, where each arm aligns to overlapping coordinates on the HIV-1 genome. This process enables the plotting of coverage and specific details of dimers by utilizing the positions and counts of these overlapping chimeras. (**B**) distribution of homodimers throughout the HIV-1 genome. The blue line plot showcases the homodimer coverage derived from ligated samples of NL4-3 infected cells along the HIV-1 genome. Arc plots exhibit discontinuous reads in non-ligated samples of NL4-3 infected cells, with

*Figure 3 continued on next page*

*Figure 3 continued*

dark red segments indicating peaks of homodimers. The lower panels depict the base pairing of homodimers in the SL1 region of the 5'-UTR and downstream of the RRE region. Color mapping indicates the log2 transformed dimer score. (**C**) Assessment of dimer self-binding using Bio-Layer Interferometry (BLI). The data presented here are from three separate experiments, offering insights into how dimers interact and bind to themselves. The dissociation constant (Kd), indicated by the mean ± standard deviation, was determined from these experiments.

The online version of this article includes the following figure supplement(s) for figure 3:

**Figure supplement 1.** Statistics of 3'–5', 5'–3', and dimer chimeras.

**Figure supplement 2.** Homodimers around HIV-1 5'-UTR.

**Figure supplement 3.** Dimer score matrices around 5'-UTR of NL4-3 and GX2005002 strain in different stages.

**Figure supplement 4.** Distribution and motif enrichment of homodimers along HIV genome.

**Figure supplement 5.** Homodimers around splicing sites.

**Figure supplement 6.** Homodimers around RRE.

**Figure supplement 7.** Validation of homodimers around RRE.

**Figure supplement 8.** Predicted structure of extended RRE.

within the HIV genome, suggesting a potential role of dimerization in splicing processes or assisting in the transport of unspliced RNA out of the nucleus.

According to a previous report that utilized CLIP-seq to analyze the RNA binding specificity of the Gag protein in both cells and virions/From a previous CLIP-seq analyzing of Gag protein's RNA binding specificity, it was observed that Gag exhibits a preference for binding to multiple elements, such as psi and RRE, in HIV infected cells (*Kutluay et al., 2014*). Interestingly, significant dimer peaks were also identified in these regions (*Figure 3—figure supplement 4A*). This suggests a potential connection between Gag binding and dimerization of the HIV genome. By plotting contact matrices derived from non-overlapping and dimeric chimeras around the RRE region, we were able to identify two distinct dimer peaks flanking the RRE region (*Figure 3—figure supplement 6A*).

To validate these novel homodimers, we synthesized extended RRE RNA fragments with two dimer sites using in vitro transcription and confirmed their ability to form a dimeric conformation through annealing and non-denaturing gel electrophoresis (*Figure 3—figure supplement 7A*), as well as Agilent Tapestation 4200 capillary electrophoresis (*Figure 3—figure supplement 7B*) and Bio-Layer Interferometry (BLI) technology (*Figure 3C*).

## Interplay between dimeric peaks and 3D genome organization

The local interaction surrounding the RRE forms an extended 'arch' structure that encompasses this element, as illustrated in *Figure 3—figure supplement 8*. This unique architecture may contribute to stabilizing the core RRE conformation, providing a structural basis for the rev-RRE interaction and HIV RNA transport. Additionally, we observed substantial local interaction signals around dimer sites in the 5'-UTR, as depicted in *Figure 3—figure supplement 2*. To provide a comprehensive overview of the local interaction landscape around dimer sites, we generated meta matrices by computing local interactions (from non-overlapping chimeras) at dimer sites and the flanking regions located 1/2 upstream and 1/2 downstream (*Figure 3—figure supplement 7*). This analysis unveiled significant local interactions around dimer sites, suggesting the involvement of stable local structures in the formation of dimers between two copies of the HIV genomes.

We observed the meta matrices around dimer sites are similar to genomic domains, which have been previously described in SARS-CoV-2 (*Zhang et al., 2021*). This observation prompted us to investigate the potential existence of genomic domains in the HIV RNA genome. Using similar methods (*Crane et al., 2015*), we calculated insulation scores for two strains of the virus in both cells and virions, as depicted in *Figure 4A*.

In this way, we identified 31, 33 HIV genomic domains in infected cells and 36, 32 domains in virions for NL4-3 and GX2005002 strains, respectively. The high correlation of insulation scores between cells and virions and consistent domain boundaries are observed (*Figure 4B*), and more importantly, the boundary strengths did not show any significant differences between cells and virions (*Figure 4C*), suggesting conserved, stable, and persistent genomic domains during the analyzed stages.

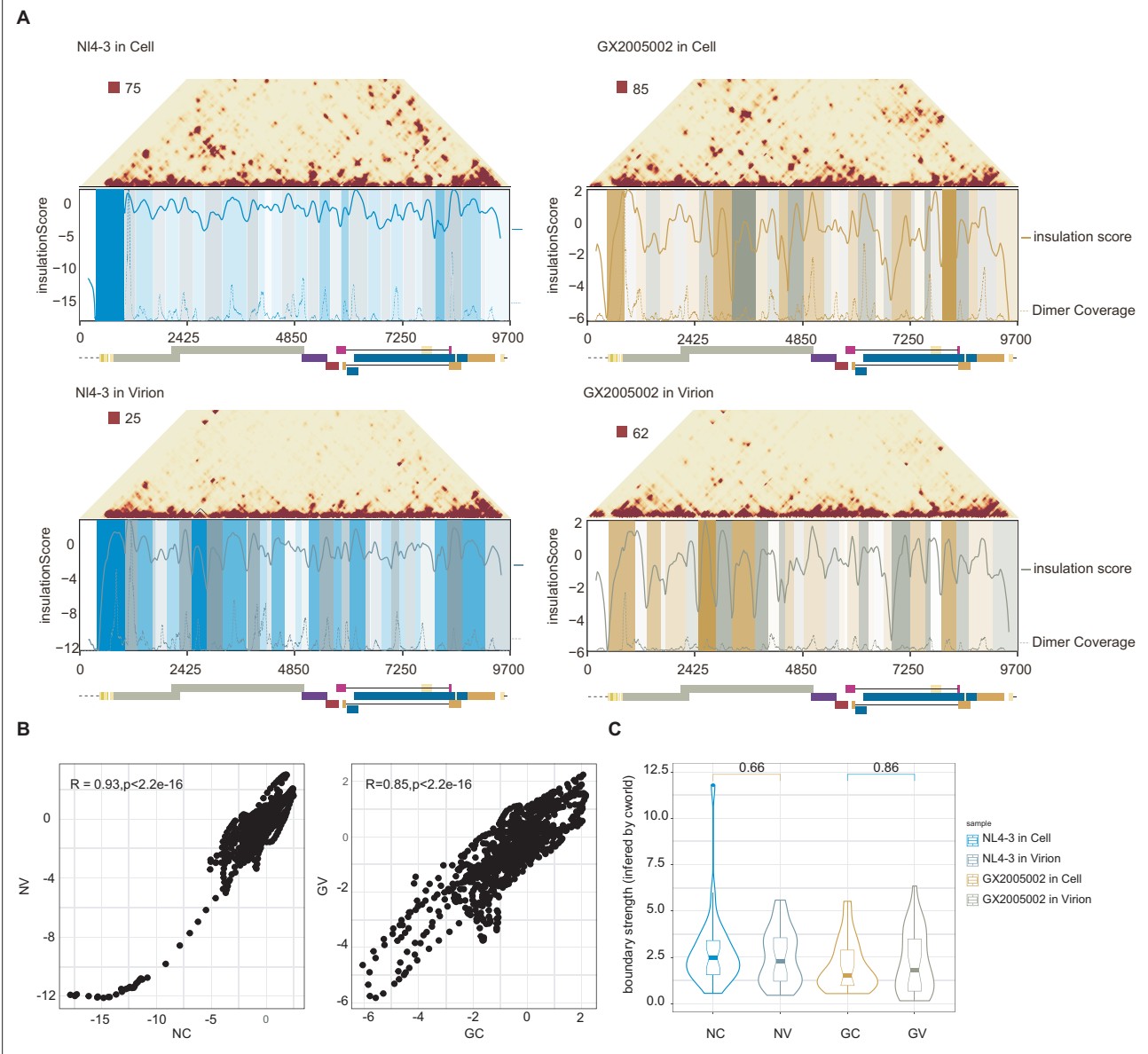

**Figure 4.** Genome domains along HIV-1 genome. (**A**) Each panel shows triangle matrix and genome domains of two strains of the virus in both cell and virion states, as calculated using C-world. The x-axis represents the genomic position, while the y-axis represents the insulation score (solid lines) and dimer coverage (dashed lines). (**B**) Correlation of insulation score in infected cells and virion for NL4-3 and GX2005002 strain. (**C**) Violin and boxplot comparing the boundary strength of the genomic domains of two strains between infected cells and virions. The boxplot displays the median, quartiles, and range of the boundary strength values for each strain in each state.

The online version of this article includes the following figure supplement(s) for figure 4:

**Figure supplement 1.** Average contact matrices around peak centers.

Interestingly, we found that dimer sites are often located within genomic domains, and overlaying dimer coverage onto domains and insulation score curves revealed a striking concordance between dimer sites and genomic domains (*Figure 4—figure supplement 1*). These findings suggest an interplay between the dimerization signals and global 3D organization of the HIV genome, providing insights into the complex mechanisms of HIV-1 replication.

# Dynamic changes in global and long-range interactions throughout HIV-1 life cycle

We noticed extensive long-range interactions between the HIV-1 genome of two strains (NL4-3 and GX2005002; *Figure 1B*, *Figure 1—figure supplement 4*).

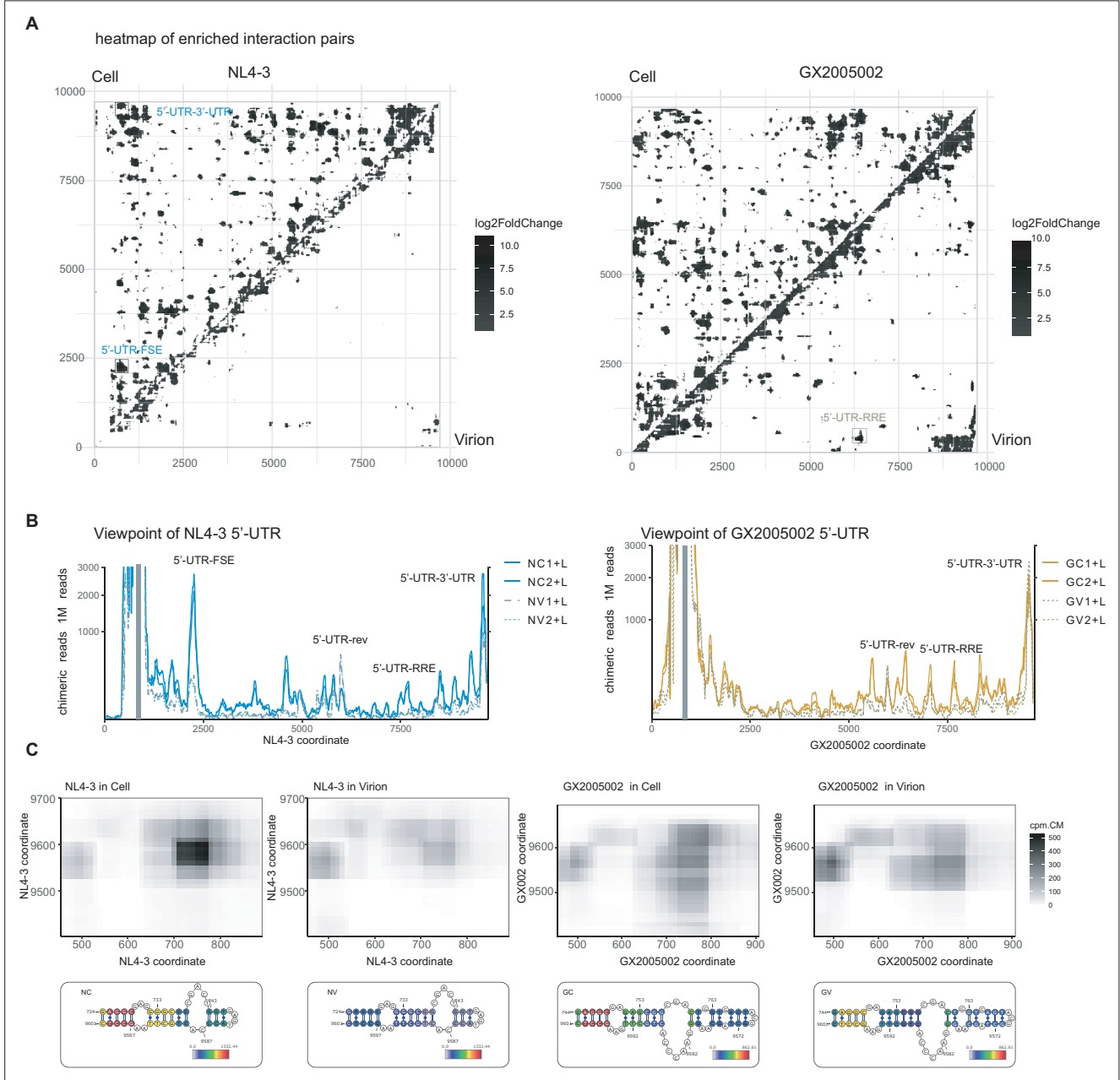

**Figure 5.** Long-range interaction in HIV-1 genome. (**A**) Heatmaps of enriched interactions obtained from NL4-3 and GX2005002 infected cells and virions. The upper diagonal shows interactions from infected cells, while the lower diagonal region displays interactions from virions. (**B**) Viewpoint lines depicts the binding positions of the 5'-UTR along the HIV-1 genome. The gray rectangles indicate the viewpoint regions. The colors of the lines represent specific samples, with samples from virions shown as dashed lines. (**C**) Contact matrices and base pairing details between 5'-UTR and 3'-UTR. The top panels display heatmaps indicating contact probability, with the color bar indicating chimeric reads per 1 M reads in each specific sample. The bottom panels show base pairing colored by base pairing scores.

The online version of this article includes the following figure supplement(s) for figure 5:

**Figure supplement 1.** Length distribution of interactions across the HIV genome.

**Figure supplement 2.** Base paring of 5'-UTR involved long range interaction.

To quantitatively analyze these interactions, we utilized DESeq2, following a similar approach as previous studies. Our results showed that both strains exhibit a substantial number of long-range interactions in infected cells, while these interactions are significantly reduced in virions (*Figure 5A*). Furthermore, histograms of enriched interaction pairs revealed a significant decrease in long-range interactions exceeding 2500nt in virions compared to cells (*Figure 5—figure supplement 1A*). The Contact probability decay curves (PS curve) also demonstrated a faster decay of genomic interactions in virions compared to cells, for both strains of the HIV-1 virus (*Figure 5—figure supplement 1B*).

Specifically, we employed viewpoint analysis to examine interactions involving the 5'-UTR. The results, depicted in *Figure 5B*, reveal multiple peaks of 5'-UTR interactions across the HIV-1 genome. Interestingly, for the majority of these peaks, the signals are lower in virions compared to cells. However, cyclization interactions (5'-UTR-3'-UTR) at the ends are still maintained in virions (*Figure 5C*). Intriguingly, we also identified multiple interaction peaks located near crucial elements, such as the frameshift element (*Figure 5—figure supplement 2A*), rev start codon (*Figure 5—figure supplement 2B*), RRE, and nuclear exporting signal (NES) (*Figure 5—figure supplement 2C*), indicating complex roles of 5'-UTR in HIV life cycle. Among these interactions, some are strain specific, such as the interaction between 5'-UTR and FSE is specific to the NL4-3 strain, while interaction between 5'-UTR and 8.5 K region (near NES) are highly enriched in GX2005002 strain (*Figure 5—figure supplement 2C*). As mentioned earlier, the majority of interactions decrease in the virion, with the notable exception of interactions in the 5'-UTR and 6 K region (near D4 splicing donor site), which significantly increase in the virion (*Figure 5—figure supplement 2B*). The enhanced interactions in virions are consistent across both strains.

These results indicate that the HIV genome undergoes systematic remodeling during virus packaging, with a general loss of long-range interactions but maintenance of specific interactions, including genome cyclization. The exploration of long-range interactions and their dynamics has provided us with an unprecedented understanding of the structural organization of the HIV genome. The significance of these interactions and their impact on the HIV life cycle and pathogenesis warrants further investigation in future research.

## Discussion

### HicapR: a reliable and efficient method for exploring structures of viral RNA genomes

In this study, we have introduced the HiCapR protocol, which combines the SPLASH method with the capture of HIV RNA-derived libraries. This integration of RNA extraction, fragmentation, proximity ligation, library construction, and subsequent capture of target RNA involved in interactions enables the investigation of various viral genomic structures. This protocol offers a streamlined and efficient approach for studying the structure and dynamics of complete HIV RNA genomes.

HiCapR uses a similar hybridization-based capture strategy as COMRADES (*Ziv et al., 2018*; *Ziv et al., 2020*) except that in HiCapR, this capture step occurs after library construction, whereas in COMRADES, it takes place prior to proximity ligation. The post-library-capture principle employed in HiCapR is a well-established approach utilized in other methods such as capture Hi-C or capture-C (*Martin et al., 2015*; *Baxter et al., 2018*; *Orlando et al., 2018*; *Chesi et al., 2019*), and it offers additional flexibility relative to previous protocols. Our results demonstrate remarkable sensitivity and reliability of the HiCapR technique in capturing low-abundance HIV RNA structure, providing an unprecedented resolution and comprehensive approach to elucidate the intricate RNA structures within the HIV-1 genome. This novel proximity ligation method transcends the limitations of traditional techniques, offering a robust framework for dissecting the complex architecture of viral RNA.

### New insight of HIV RNA local and long-range interactions

The 5'-UTR is crucial for various stages in the HIV life cycle. Our analysis of its structures across different strains and conditions revealed consistent canonical stem-loops and interactions, despite sequence variations and alternative conformations, highlighting the structural robustness? stability and functional significance of the HIV 5'-UTR.

In addition to its local folding, the 5'-UTR engages in extensive and dynamic long-range interactions, notably with the 3'-UTR, suggesting genome cyclization akin to other viruses (*Figure 4*). This

phenomenon, also observed in Zika, influenza, and SARS-CoV-2 (*Miyamoto and Noda, 2020*; *Ziv et al., 2018*; *Ziv et al., 2020*; *Zhang et al., 2021*), hints at broader biological implications warranting further exploration. Additionally, the 5'-UTR interacts with key import elements in the HIV genome, such as NES, RRE, and the rev start codon. Given the Rev protein's roles in RNA transport and nuclear export (*Li and De Clercq, 2016*), the 5'-UTR may also contribute to these processes.

Previous studies have highlighted folding principles of Zika, SARS-COV-2 coronavirus, and influenza virus, unraveling the intricate mechanisms underlying viral replication, pathogenesis, and host interactions (*Miyamoto and Noda, 2020*; *Ziv et al., 2018*; *Cao et al., 2021*; *Zhang et al., 2021*; *Dadonaite et al., 2019*). One of the key findings from this study is that the HIV-1 genome is organized in a complex three-dimensional structure that facilitates long-range interactions between distant regions of the genome. This study shed light on the dynamics of long-range interactions between the HIV-1 genome of two strains, NL4-3 and GX2005002. These interactions are largely reduced in virions compared to cells, suggesting a critical role in virus assembly and release.

Previous studies has elucidated the folding principles of various viruses like Zika, SARS-CoV-2, and influenza, shedding light on viral replication, pathogenesis, and host interactions (*Miyamoto and Noda, 2020*; *Ziv et al., 2018*; *Cao et al., 2021*; *Zhang et al., 2021*; *Dadonaite et al., 2019*). Our study revealed that the HIV-1 genome's intricate three-dimensional organization enables long-range interactions.

The contrasting loss of long-range interactions in the HIV-1 genome compared to SARS-CoV-2 (*Zhang et al., 2021*) suggests distinct folding processes between these viruses. This disparity underscores the unique characteristics of each virus and their genomic structures. Speculations indicate that the HIV-1 genome may adopt a rod-like structure within viral particles, unlike the spherical compression seen in SARS-CoV-2 (*Yao et al., 2020*). Further investigation is needed to unravel the mechanisms and biological significance of this compression in the HIV-1 genome.

## Emerging insights into HIV-1 dimerization through newly identified dimer sites

Our detailed analysis has consistently identified approximately 20 candidate dimer peaks both within and beyond the 5'-untranslated region (5'-UTR) across the NL4-3 and GX2005002 strains of HIV. Intriguingly, these peaks show a notable enrichment in the vicinity of splicing sites, frequently featuring a sequence motif that bears a strong resemblance to the RNA binding signature of the Serine/Arginine-Rich Splicing Factors (SRSF). This enrichment pattern and the presence of the SRSF-like motif at these sites suggest a potential regulatory role of these dimers in the complex landscape of HIV RNA splicing. The pervasive presence of homodimers flanking almost every splicing site introduces the intriguing hypothesis that these dimers could be actively participating in the regulation of alternative splicing or facilitating the export of unspliced genomic RNA from the nucleus to the cytoplasm. This hypothesis warrants rigorous investigation in subsequent studies to elucidate the mechanistic underpinnings of these observations. Collectively, these findings hint at a significant interplay between RNA dimerization and splicing processes within the HIV genome, which could have profound implications for our understanding of viral gene expression and replication strategies.

Moreover, significant dimer signals were observed in the RRE flanking sequences, adding complexity to its functional role, potentially impacting nucleocytoplasmic transport, Gag binding, and virus packaging processes. Given previous studies highlighting Gag binding to RRE and the crucial role of RRE in anchoring Gag synthesis (*Emery and Swanstrom, 2021*; *Kutluay et al., 2014*; *Becker and Sherer, 2017*), it is further hypothesized that HIV-1 homodimers may play a role in HIV splicing or assist in transporting unspliced full-length RNA out of the nucleus for genome packaging.

Interestingly, the observation that candidate dimer sites are located within genomic domains suggests that dimerization may be influenced by the local RNA folding environment (*Figure 3—figure supplement 7*). It is possible that the dense local interaction around these sites may facilitate dimerization by bringing the two regions of the genome into close proximity. Alternatively, dimerization may play a role in shaping the local chromatin environment by promoting even initiating the formation of genomic domains. Further research is needed to fully understand the relationship between RNA genomic domains and dimer sites in the HIV-1 genome. However, the identification of these candidate dimer sites within genomic domains provides a starting point for investigating the role of local genome RNA interactions in HIV-1 replication and dimerization. These findings may have implications

for the development of novel antiviral therapies that target the dimerization process and may provide new avenues for future research in this field.

In summary, our study provides a comprehensive analysis of the HIV-1 genome using reliable HicapR, revealing potential dimer sites and long-range interactions, particularly within the 5'-UTR as well as persistent structure domains? These findings significantly advance our understanding of the structure, dynamics and robustness of HIV RNA and their involvement in splicing, assembly and packaging processes, potentially leading to the development of novel antiviral therapies.

## Methods

### Cell culture and virus infection

The MT4 cell lines (RRID:CVCL_2632) were cultured under specific conditions: RPMI 1640 medium (Gibco, USA) supplemented with 10% fetal calf serum (Gibco, USA), 100 U/ml penicillin, and 100 µg/ml streptomycin. MT-4 cells are human acute lymphoblastic leukemia cells derived from peripheral blood leukocytes of a 50-year-old Japanese male adult T-cell leukemia patient, established through co-culture with umbilical cord lymphocytes from a male infant. The cells are lymphoblastic-like, round, and grow in suspension. MT-4 cells express CD4 receptors, which are key receptors for the invasion of viruses such as HIV, making MT-4 cells highly sensitive to HIV. The MT-4 cell line we use was kindly donated by Professor Lu Shan from the University of Massachusetts Medical School. The identity of the cell line has been verified by STR profiling, and the results show that the STR profile of this MT-4 cell line is consistent with the standard reference profile. Besides, the cell line tested negative for mycoplasma contamination. The cells were maintained at a temperature of 37 °C with 5% $CO_2$ and saturated humidity. To initiate infection, cells were exposed to HIV-1 NL4-3 (HIV-1 strain of subtype B) or HIV-1 GX2005002 (HIV-1 primary strain of CRF01_AE which is one of the main strain in China, accession: GU564222) at a multiplicity of infection (MOI) of 0.15. Concurrently, parallel control groups consisting of uninfected cells were also established. Both the cells and the cell supernatant were collected after 48 hr post infection for subsequent experiments.

### HicapR method

First, extracted crosslinked RNA was treated as in simplified SPLASH protocol (*Zhang et al., 2021*).

Briefly, 500 ng of each sample was fragmented using RNase III (Ambion) in a 20 µl mixture for 10 min at 37 °C. The fragmented RNA was then purified using 40 µl of MagicPure RNA Beads (TransGen). Each RNA sample was subsequently divided into two halves, with one half used for proximity ligation and crosslink reversal (C, V samples). The proximity ligation process was then carried out using the following conditions: 200 ng of fragmented RNA, 1 unit/µl RNA ligase 1 (New England Biolabs), 1×RNA ligase buffer, 1 mM ATP, 1 unit/µl Superase-in (Invitrogen), with a final volume of 200 µl. The reactions were incubated for 16 hr at 16 °C and were stopped by cleaning with the miRNeasy kit (QIAGEN). To reverse the crosslinking, the RNA was irradiated on ice with 254 nm UltraViolet C radiation for 5 min using a CL-1000 crosslinker (UVP). For the non-ligated controls, crosslink reversal was performed immediately after crosslinking, without proximity ligation (these controls were labeled as '-L').

The proximity ligated RNA was then subjected to library construction using SMARTer Stranded Total RNA-Seq Kit v2 (Clonetech).

The cDNA libraries were enriched for HIV fragment using the TargetSeq One Hyb & Wash kit (igenetech) with the T548XV1 probe panel, designed based on HIV genome. The probe sequences are provided as *Figure 1—source data 1*.

The virus inoculation, crosslinking, and HiCapR were performed on two independent replicates.

### Data preprocessing and chimeric reads identification

The data preprocessing and identification of chimeric reads were conducted as previously described. The reference genome used for NL4-3 strain was a combination of NL4-3 genome sequence (https://www.ncbi.nlm.nih.gov/nuccore/AF003887) and human spliced mRNAs and noncoding RNAs described in *Travis et al., 2014*, while for GX2005002 strain, the reference genome used was a combination of GX2005002 genome sequence (https://www.ncbi.nlm.nih.gov/nuccore/KP178420) and human spliced mRNAs and noncoding RNAs as above.

First, we used pear to merge the overlapped reads:

```
pear -e -j 32 f sample_R2.fastq.gz -r sample_R1.fastq.gz -o Sample.PEAR
```

note that for Clonetech 634413, the sense strand of RNAs is in R2 read.

Then, we used fastp to filter reads with low quality and cut adapters:

```
fastp -i sample.PEAR.fastq -o sample.PEAR.fastp.fastq -h fastp.html -w 4 a AGATCGGAAGAG
CGTCG
```

And then, chimeric reads are detected using hyb pipeline with default setting: hyb detect in = sample.PEAR.fastp.fastq db=$DB qc = none

## Chimeras for RNA-RNA interaction and dimer identification

Homodimers were identified in accordance with the methods previously reported in the literature, with a more stringent filter (*Gabryelska et al., 2022*). Specifically, we utilized the hub pipeline and filtered chimeras in.hyb files. To calculate the degree of overlap between the two arms of chimeric reads, we used the formula $L = 1 + \min(e1, e2) - \max(s1, s2)$, where $e1$ and $e2$ represent the ends of each arm of chimeric reads, while $s1$ and $s2$ represent their starts. To ensure that possible circular RNAs, which may be produced by single RNA end-to-end ligation, are filtered out, we applied the following condition: ($e1 < e2$) OR ($s1 < s2$). A schematic diagram of dimer chimeras is provided in *Figure 1— figure supplement 1*.

We defined the dimer range as the maximum value between $s1$ and $s2$, and the minimum value between $e1$ and $e2$. Based on this range, we further calculated the coverage of dimer chimeras.

On the other hand, only non-overlapping chimeras (with $L < 0$) were included for RNA-RNA interaction contact matrices to reduce the possibility that the chimeras called for RNA-RNA interactions come from homodimers.

## Dimer score and dimer base pairing

The dimer score (DS) was defined as the number of chimeric reads that supported a potential dimer base-pairing event. To visualize each particular candidate dimer, we used the hybrid-min command in the RNA fold package to in-silico hybridize two homodimers. The paired bases were then colored using the aforementioned dimer score.

## Local structure folding and MDS plot

Initial local structure folding was performed similarly to a previous report (*Ziv et al., 2018*). Base pairing scores were first calculated using comradesMakeConstraints function in COMRADES package (https://github.com/gkudla/comrades; *gkudla, 2020*):

```
comradesMakeConstraints -i sample_rm_overlap.hyb -f Genome_fasta -b 1 -e genome_length
```

we folded the RNA structures 1000 times using the COMRADES manual. Subsequently, they utilized multidimensional scaling to calculate the distances between RNA structures.

## Calculation of base pairing probability for local structure

For a particular cluster in the MDS plot, we construct a base pairing probability matrix to reveal consensus stems in these structures, where the probability of base pairing for each base pair is determined by calculating the proportion of structures that support the base pair among all folded structures.

## Dimer validation

### RNA preparation

5'-UTR and RRE DNA fragments were amplified by RT-PCR using RNA from infected cells, with primers listed in *Supplementary file 3*. Purified NL4-3 and GX2005002 5'-UTR and extended RRE PCR products (200 ng) were utilized as templates for RNA in vitro transcription with T7 RNA polymerase (Vazyme), following manufacturer's protocol. The reaction was then incubated at 37 °C for 16 hr, followed by DNase I treatment for 30 min at 37 °C. The RNA was subsequently purified using MagicPure RNA Beads (TransGen) through gel purification.

## Native agarose gel electrophoresis

RNA (600 ng) was heated to 95 °C and then slowly cooled to room temperature in either high-salt buffer (50 mM Tris-HCl pH 7.5, 140 mM KCl, 10 mM NaCl) or low-salt buffer (high-salt buffer diluted 1:10 with water). Samples were loaded with native loading dye (0.17% Bromophenol Blue and 40% (vol/vol) sucrose) on 2% agarose gel prepared with 1×tris-borate magnesium (TBM) buffer (89 mM Tris base, 89 mM boric acid and 2 mM MgCl2) and fractionated at 100 V for 85 min at room temperature. RNA in the PAGE gel was visualized using GelRed nucleic acid gel stain (Thomas scientific).

## Agilent 4200 TapeStation capillary electrophoresis

We prepared the RNA samples as described above and loaded them onto the TapeStation RNA screentape without the heating and denaturing step. Subsequently, we initiated the device for electrophoresis analysis, which ran automatically without the need for manual intervention. After the electrophoresis analysis was completed, the system generated electropherograms and relevant data regarding the RNA samples, including their size distribution and concentration. This method allowed us to quickly and accurately evaluate the conformational features of the RNA.

## Biomolecular binding kinetics assays

Biomolecular Binding Kinetics Assays were performed using the Octet R8 Platform (sartorius) following a standardized protocol. The assay involved the preparation of analytes at varying concentrations, real-time monitoring of association and dissociation phases, and subsequent data analysis to determine kinetic parameters kd. Quality control measures were implemented to ensure the reliability and reproducibility of the binding kinetics measurements.

## Code availability

Custom code for the analysis performed in this study is publicly available via biocode at https://ngdc.cncb.ac.cn/biocode/tools/BT007456.

---

## Additional information

### Funding

No external funding was received for this work.

### Author contributions

Yan Zhang, Conceptualization, Formal analysis, Writing – original draft; Jingwan Han, Resources, Data curation; Xie Dejian, Formal analysis; Wenlong Shen, Investigation; Ping Li, Methodology; Jian You Lau, Validation; Jingyun Li, Writing – review and editing; Lin Li, Conceptualization, Investigation; Grzegorz Kudla, Conceptualization, Formal analysis, Writing – review and editing; Zhihu Zhao, Conceptualization, Supervision

### Author ORCIDs

Yan Zhang (iD) https://orcid.org/0000-0003-4006-663X
Xie Dejian (iD) https://orcid.org/0009-0000-3025-3568
Wenlong Shen (iD) https://orcid.org/0000-0002-2293-5590
Lin Li (iD) https://orcid.org/0000-0001-6124-5237
Grzegorz Kudla (iD) https://orcid.org/0000-0002-7924-2744
Zhihu Zhao (iD) https://orcid.org/0000-0001-7081-8914

Reviewer #1 (Public review): https://doi.org/10.7554/eLife.102550.3.sa1
Reviewer #2 (Public review): https://doi.org/10.7554/eLife.102550.3.sa2
Author response https://doi.org/10.7554/eLife.102550.3.sa3

## Additional files

### Supplementary files

Supplementary file 1. Mapping statistics.

Supplementary file 2. Homodimer peaks information.

Supplementary file 3. Primers used for amplifying 5'-UTR and extended RRE.

MDAR checklist

### Data availability

The raw sequence data reported in this paper have been deposited in the Genome Sequence Archive (*Chen et al., 2021*) in National Genomics Data Center (*CNCB-NGDC Members and Partners, 2022*), China National Center for Bioinformation/Beijing Institute of Genomics, Chinese Academy of Sciences (GSA: CRA016024) and are publicly accessible at https://ngdc.cncb.ac.cn/gsa. Reused Gag CLIP-seq data were retrieved from the GEO database with accession number GSE61508.

The following dataset was generated:

| Author(s) | Year | Dataset title | Dataset URL | Database and Identifier |
|---|---|---|---|---|
| Zhao Z | 2025 | Mapping HIV-1 RNA Structure, Homodimers, and Long-Range Interactions by HiCapR | https://ngdc.cncb. ac.cn/gsa/search? searchTerm= CRA016024 | Genome Sequence Archive, CRA016024 |

The following previously published dataset was used:

| Author(s) | Year | Dataset title | Dataset URL | Database and Identifier |
|---|---|---|---|---|
| Kutluay SB, Zang T, Blanco-Melo D, Powell C | 2014 | 293T cells transfected with HIV-1 proviral plasmids | https://www.ncbi. nlm.nih.gov/geo/ query/acc.cgi?acc= gse61508 | NCBI Gene Expression Omnibus, GSE61508 |

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
