## [Editor Report · eLife Assessment]

This manuscript focuses on the identification of RNA crosslinks within the HIV RNA genome under different conditions i.e. in infected cells and in virions using a new method called HiCapR. These cross-links reveal long-range interactions that can be used to determine the structural arrangement of the viral RNA, providing **valuable** data that show differences in the genomic organization in different conditions. The data analysis, however, is **incomplete** and based on extensive computational analysis from a limited number of datasets.

---

## [Referee Report · Reviewer #1 (Public review)]

This paper focuses on secondary structure and homodimers in the HIV genome. The authors introduce a new method called HiCapR which reveals secondary structure, homodimer, and long-range interactions in the HIV genome. The experimental design and data analysis are well-documented and statistically sound.

Comments on revisions:

The authors have addressed key questions and highlighted the advantages of HiCapR.

---

## [Referee Report · Reviewer #2 (Public review)]

Summary:

In the manuscript "Mapping HIV-1 RNA Structure, Homodimers, Long-Range Interactions and 1 persistent domains by HiCapR" Zhang et al report results from an omics-type approach to mapping RNA crosslinks within the HIV RNA genome under different conditions i.e. in infected cells and in virions. Reportedly, they used a previously published method which, in the present case, was improved for application to RNAs of low abundance.

Their claims include the detection of numerous long-range interactions, some of which differ between cellular and virion RNA. Further claims concern the detection and analysis of homodimers.

Strengths:

(1) The method developed here works with extremely little viral RNA input and allows for the comparison of RNA from infected cells versus virions.

(2) The findings, if validated properly, are certainly interesting to the community.

Weaknesses:

(1) On the communication level, the present version of the manuscript suffers from a number of shortcomings. I may be insufficiently familiar with habits in this community, but for RNA afficionados just a little bit outside of the viral-RNA-X-link community, the original method (reference 22) and the presumed improvement here are far too little explained, namely in something like three lines (98-100). This is not at all conducive to further reading.

(2) Experimentally, the manuscript seems to be based on a single biological replicate, so there is strong concern about reproducibility.

(3) The authors perform an extensive computational analysis from a limited number of datasets, which are in thorough need of experimental validation

Comments on revisions:

The authors have made cosmetic changes with regards to the problems I raised. 1 - Reproducibilty: the rebuttal letter says there are now 3 replicates, but there is only data for 2 in the supplement. The generation of biological replicates needs to be precisely stated, e.g. taken on different days, from separate cultures, or from neighbouring dishes on the same day etc. I think, the manuscript would greatly benefit from the comparison of at least 3 replicates that were not generated on the same day. Given that the authors report a r2 of 0.99 between the sets they have, this seems quite plausible.

The validation of the dimerisation sites is marginally better, but the authors should read up on significant digits and how precise Kd values can be determined.

The authors state that they want to make several of the experimeriments that would address my issues in the future in the context of another study. I find that disappointing, and correspondingly the present datasets insufficient for further endorsement.

---

## [Author Response]

The following is the authors’ response to the original reviews.

**Public Reviews:**

**Reviewer #1 (Public review)**
This paper focuses on secondary structure and homodimers in the HIV genome. The authors introduce a new method called HiCapR which reveals secondary structure, homodimer, and long-range interactions in the HIV genome. The experimental design and data analysis are well-documented and statistically sound. However, the manuscript could be further improved in the following aspects.Major comments:(1) Please give the full name of an abbreviation the first time it appears in the paper, for example, in L37, "5' UTR" "RRE".

Thank you for your suggestion. We have added the full name of these abbreviations.

(2) The introduction could be strengthened by discussing the limitations of existing methods for studying HIV RNA structures and interactions and highlighting the specific advantages of the HiCapR method.

Thank you for your insightful suggestion. We have modifed sentences in the introduction section (line 66 -line 71, line 80-line 81 in the revised manuscript).

(3) Please reorganize Results Part 1.

Thank you for your advice. We have reorganized results part 1. We hope the revision provides a logical flow and clarity to the results, making it easier for readers to follow the progression of the study and the significance of the findings regarding to the HiCapR method.

(4) Is there any reason that the authors mention "genome structure of SARS-CoV-2" in L95?

Thank you for your insightful question. We have deleted this sentence in the revised paper.

Initially, the mention of our previous work on SARS-CoV-2 serves two purposes: firstly, to demonstrate our capability to perform proximity ligation assays on viral samples; and secondly, to underscore the necessity of the hybridization step, which is particularly relevant for the study of HIV.

Unlike SARS-CoV-2, which is highly abundant in infected cells and does not require post-library hybridization, HIV-1 presents a unique challenge due to its typically low viral RNA input within cells. The simplified SPLASH protocol, while effective for more abundant viral RNAs, does not provide the necessary coverage for high-resolution analysis when applied directly to HIV samples.

Now, we have deleted this sentence according to your comments, and discuss the technical difference elsewhere.

(5) L102: Please clarify the purpose of comparing "NL4-3" and "GX2005002." Additionally, could you explain what NL4-3 and GX2005002 are? The connection between NL4-3, GX2005002, and HIV appears to be missing.

Thank you for your question, and we apologize for the misleading. "NL4-3" and "GX2005002" are two distinct HIV-1 strains that exhibit different prevalence patterns in various geographical regions. The NL4-3 strain is a well-characterized laboratory strain that is widely used in HIV research and is representative of the HIV-1 subtype B, which is highly prevalent in Europe and the Americas. On the other hand, GX2005002 is a primary isolate of the CRF01_AE subtype, which is one of the most prevalent strains in Southeast Asia, particularly in China.

The reason for comparing these two strains in our study is twofold. Firstly, it allows us to assess the applicability and versatility of our HiCapR method across different HIV-1 strains that may have distinct genetic and structural features. This is crucial for understanding the potential broad utility of our method in studying various HIV-1 strains globally. Secondly, by comparing these strains, we can begin to elucidate any strain-specific differences in RNA structure, homodimer formation, and long-range interactions, which may have implications for viral pathogenesis, transmission, and response to therapeutic interventions.

The connection between NL4-3, GX2005002, and HIV lies in their representation of different subtypes of the HIV-1 virus, which exhibit genetic diversity and are associated with different geographical distributions. This diversity is epidemiologically and clinically relevant, as it may be associated with different pathogenesis and resistance mechanisms, and might has implications for vaccine development and treatment strategies.

(6) Figure 1A is not able to clearly present the innovation point of HiCapR.

Thank you for your comment. We have revised this figure to more clearly illustrate the steps and principles of the post-library capture process using HIV pooled probes hybridization and streptavidin pull down to enrich HIV RNA-derived chimeras.

(7) Please compare the contact metrics detected by HiCapR and current techniques like SHAPE on the local interactions to assess the accuracy of HiCapR in capturing local RNA interactions relative to established methods.

Thank you for your request to compare the contact metrics detected by HiCapR and current techniques like SHAPE on local interactions to assess the accuracy of HiCapR in capturing local RNA interactions relative to established methods.

In this study, HiCapR has demonstrated its ability to identify key structural elements within the HIV genome, including TAR, polyA, SL1, SL2, and SL3, as well as the polyA-SL1 in the monomeric conformation. These elements are crucial for understanding the local RNA structures involved in HIV replication and pathogenesis. By visualizing the base pairing probability as a heatmap, we have identified the most stable base pairs in the 5’ UTR of HIV, which is consistent across both NL4-3 and GX2005002 strains (Figure 2D). This consistency suggests robustness in the overall structure despite sequence variations and alternative RNA conformations, indicating a high level of agreement between HiCapR and SHAPE methods in detecting local interactions.

Furthermore, HiCapR not only confirms the presence of known structural elements but also reveals alternative conformations of the 5'UTR that support the alternative conformations found in SHAPE analysis. This additional layer of information provides a more comprehensive view of the RNA structures, highlighting HiCapR's ability to capture local RNA interactions with a high degree of accuracy comparable to established methods like SHAPE.

(8) The paper needs further language editing.

We have thoroughly revised the paper. We hope it’s improved significantly.

**Reviewer #2 (Public review):**
Summary:In the manuscript "Mapping HIV-1 RNA Structure, Homodimers, Long-Range Interactions and 1 persistent domains by HiCapR" Zhang et al report results from an omics-type approach to mapping RNA crosslinks within the HIV RNA genome under different conditions i.e. in infected cells and in virions. Reportedly, they used a previously published method which, in the present case, was improved for application to RNAs of low abundance.Their claims include the detection of numerous long-range interactions, some of which differ between cellular and virion RNA. Further claims concern the detection and analysis of homodimers.Strengths:(1) The method developed here works with extremely little viral RNA input and allows for the comparison of RNA from infected cells versus virions.(2) The findings, if validated properly, are certainly interesting to the community.

Thank you for your comprehensive review and insightful comments on our manuscript. We appreciate your recognition of the strengths of our HiCapR method and the potential interest of our findings to the scientific community.

Weaknesses:(1) On the communication level, the present version of the manuscript suffers from a number of shortcomings. I may be insufficiently familiar with habits in this community, but for RNA afficionados just a little bit outside of the viral-RNA-X-link community, the original method (reference 22) and the presumed improvement here are far too little explained, namely in something like three lines (98-100). This is not at all conducive to further reading.

Thank you for your feedback on the clarity of our manuscript, particularly regarding the explanation of the HiCapR method and its improvements over the original method mentioned in reference 22

In response to your feedback, we expand on the description of the HiCapR method in the revised manuscript to ensure that it is accessible to a broader audience. We will provide a more thorough comparison between HiCapR and the original method, detailing the specific improvements and how they enable the analysis of low-abundance viral RNAs like HIV. This will include:

Post-library Hybridization: Unlike the original method, HiCapR incorporates a post-library hybridization step. This innovation allows for the capture of target RNA involved in interactions after library construction, offering additional flexibility and enhancing the resolution of the analysis.

Enhanced Sensitivity: HiCapR has been optimized to work with extremely low viral RNA input, which is a significant advancement over the original method. This is crucial for studying viruses like HIV, where obtaining high quantities of viral RNA can be challenging. As a matter of fact,

(2) Experimentally, the manuscript seems to be based on a single biological replicate, so there is strong concern about reproducibility.

Thank you for raising the issue of reproducibility in our study. We understand the importance of experimental replication in ensuring the reliability of our findings. In response to your concern, we would like to provide the following clarification and additional details regarding the reproducibility of our HiCapR experiments:

Replicates in HiCapR Experiments: All ligation and control samples in our HiCapR experiments were performed in three biological replicates. This was done to ensure the high reproducibility of our results. The high degree of correlation (r > 0.99) between these replicates underscores the reliability of our findings.

Dimer Validation Experiments: To validate the dimer formation of RRE and 5’-UTR, we employed multiple independent methods, including Native agarose gel electrophoresis, Agilent 4200 TapeStation Capillary electrophoresis, and Biomolecular Binding Kinetics Assays. These methods provide complementary perspectives on the dimer formation, enhancing the robustness of our validation process. The data presented in Figure 3C and Supplementary figure S12 are representative results from these experiments, which consistently support our findings on dimer formation.

Agreement Between Cellular and Virion RNA: Our study also demonstrates a significant similarity between virions in the supernatant and infected cells from the same viral strain, as shown in Supplementary Figure S3. This consistency further validates the reproducibility and reliability of our HiCapR method in capturing RNA structures and interactions under different conditions.

Consistency across two strains: Our study includes a comprehensive analysis of two distinct HIV-1 strains, NL4-3 and GX2005002, which are prevalent in Europe and Southeast Asia, respectively. The consistency in our findings across these strains serves as a strong indicator of the reproducibility and general applicability of our HiCapR method. Specifically, presence of key structural elements such as TAR, polyA, SL1, SL2, and SL3 in both NL4-3 and GX2005002 strains, suggests a robust structural framework that is conserved across different strains, despite sequence variations. Additionally, our study reveals approximately 20 candidate dimer peaks conserved between the NL4-3 and GX2005002 strains along the genome. The conservation of these dimer peaks across strains indicates a reproducible pattern of dimerization.

(3) The authors perform an extensive computational analysis from a limited number of datasets, which are in thorough need of experimental validation

Thank you for your comment.

In response to your concern, we would like to clarify that while our manuscript does present an extensive computational analysis, we have also conducted a series of experiments. Specifically, we have validated dimer formation using multiple independent methods (afore discussed).

Given the time-consuming nature of additional experiments, we have chosen to share the HiCapR data with the community in a timely manner. This approach allows for more immediate communication and evaluation of the data on HIV structure, which we believe is valuable for advancing the field.

We are committed to further investigating the functional implications of our structural findings. We plan to conduct more experiments to explore the functional linking between the structural insights of HIV, which will help to deepen our understanding of the virus's replication and potential antiviral strategies.

**Recommendations for the authors:**

**Reviewer #1 (Recommendations for the authors):**
I suggest a major revision of the manuscript.Minor comments:(1) The article lacks consistency in its presentation. The expression of the proper noun is wrong in the paper. For example, (a) L89, "RNA:RNA interaction" →RNA-RNA interaction; (b) L431, "SARS-COV-2" → SARS-CoV-2;

We are sorry for the inconsistency. We have corrected the mistakes.

(2) "We identified dimers based on the methodology described in23." is not a complete sentence.

Thank you for your insightful comment. We have revised the sentence to provide a complete and clear description of our methodology. The revised sentence is as follows: "Homodimers were identified in accordance with the methods previously reported in the literature."

**Reviewer #2 (Recommendations for the authors):**
(1) The authors perform an extensive computational analysis from a limited number of datasets, which are in thorough need of experimental validation. There is a single series on in vitro validation of the interaction of an homodimerization site, described in five lines (278-283) plus the Figure panel 3c with a very brief legend, and an extremely minimalist Figure S12. The panel to Figure 3c contains Kd values which have not been assessed for significant digits.

Thank you for your constructive feedback on our manuscript.

We acknowledge that our computational analysis is based on a limited number of datasets. Due to the initial exploratory nature of our study and the logistical challenges of generating additional datasets, we have focused on in-depth analysis of the available data. We are currently working on further validating our findings and are committed to publishing these results in a follow-up study.

Regarding Experimental Validation:

We agree that the initial description of our in vitro validation of the homodimerization site was concise. To address this, we have expanded the description of our experimental procedures. Specifically, we have detailed the methods used for the in vitro transcription, the preparation of RNA samples, and the use of the Octet R8 platform for biomolecular binding kinetics assays.

For the Kd values presented in Figure 3c. We have now included standard error of the mean and have defined the significant digits in the figure legend. This revision provides a more accurate representation of the binding affinities.

(2) As a further example to be experimentally validated, splice sites are discussed after lines 354, for which unsophisticated validation techniques such as targeted RT-PCR are widely accepted.

In response to your comment, we would like to clarify that the splice sites mentioned in our study are well-established and widely recognized in the literature. They have been previously characterized and are considered canonical within the HIV research community. Given their established nature, we have relied on this foundational knowledge in our analysis.

However, we concur with the importance of validating the regulatory role of homodimers in splicing, which is a significant aspect of HIV biology. While we have provided evidence for the presence of these homodimers and their potential implications for splicing, we acknowledge the need for further functional studies to elucidate their mechanistic role.

Due to the scope and length constraints of the current manuscript, we have chosen to focus on the structural and interaction analyses provided by HiCapR. The functional validation of these homodimers and their impact on splicing will be pursued in subsequent studies, which we plan to initiate promptly. We believe that a dedicated follow-up study will allow for a more in-depth exploration of this complex and important aspect of HIV gene regulation.

We are committed to advancing our understanding of the functional significance of these homodimers in the context of HIV splicing and will ensure that this line of investigation is thoroughly addressed in our future work.

Thank you again for your valuable feedback. We look forward to contributing further to the field with our ongoing research.